# Mechanisms of Corneal Nerve Regeneration: Examining Molecular Regulators

**DOI:** 10.3390/cells14171322

**Published:** 2025-08-27

**Authors:** Bianca Bigit, Victor H. Guaiquil, Ali R. Djalilian, Mark I. Rosenblatt

**Affiliations:** Department of Ophthalmology and Visual Sciences, Illinois Eye and Ear Infirmary, College of Medicine, University of Illinois at Chicago, Chicago, IL 60612, USA; bbigit2@uic.edu (B.B.); vguaiqui@uic.edu (V.H.G.)

**Keywords:** corneal, nerve, regeneration, anatomical, repair, neurotrophic, regulators

## Abstract

Corneal nerve integrity is vital for maintaining ocular surface health and visual clarity, but damage from injury or disease can lead to pain, persistent epithelial defects, and even vision loss. A deeper understanding of how corneal nerves regenerate at the molecular level is key to developing therapies that restore both anatomical structure and function. In this review, we bring together current insights into the pathways that drive corneal nerve repair after injury. We outline the major signaling pathways that promote neuronal survival, axon extension, and nerve–epithelial interactions, along with evolving research around novel modulators that could improve repair outcomes. Although advances in imaging and molecular therapies have led to significant progress in promoting nerve regrowth, functional sensory recovery often lags. This gap in recovery emphasizes the need for research approaches that align anatomical restoration with sensory function. In this review, we aim to clarify the mechanisms underlying corneal nerve regeneration (and their intersections) and identify opportunities for improving patient outcomes.

## 1. Introduction

The cornea is the most densely innervated tissue in the body and detects mechanical, chemical, and thermal stimuli, ensuring nerve and epithelial cell integrity while preserving corneal clarity [1]. As the most anterior part of the cornea, sensory nerves are essential for ocular sensation, blinking, tear production, and maintaining epithelial integrity through trophic support [2,3]. Corneal sensory neurons originate from the ophthalmic branch of the trigeminal ganglia (TG) and—with an extensive network of nerve terminals in the corneal epithelium—relay afferent sensory information from the ocular surface to the brain to maintain corneal homeostasis [4,5,6]. A defining feature of animal models’ corneal nerve architecture is its highly organized whorl-like pattern. Sensory nerve fibers projecting from the ophthalmic branch of the TG enter radially in the stroma. As nerve fibers extend more centrally during development, the fibers curve and spiral, creating a vortex pattern just under the subbasal plexus beneath the epithelium [7,8]. This patterning allows for the uniform distribution of nerve terminals in the epithelium, an essential feature for maintaining corneal homeostasis. Recovery of the whorl-like pattern after injury is a vital component for regaining corneal function, and changes observed in the whorl could indicate neuropathy or disease progression [9,10,11]. In humans, the central whorl is often less morphologically distinct, with more of a web-like network, rather than a sharply defined vortex as usually seen in mice or macaques [3].

Corneal nerve damage can occur from a diverse range of insults: refractive surgery, infections, chemical injury, diabetes, or autoimmune diseases [1]. As a result, abnormal corneal nerve function and density can leave patients with a lack of sensation or, in some cases, chronic pain [1,6,12]. Corneal nerve repair involves a coordinated sequence of molecular events from injury to resolution, and understanding this sequence is critical for maximizing patient outcomes and developing therapeutics. Corneal nerves exist intimately with their surrounding environment, maintaining homeostasis with the help of tropic support of epithelial cells and the extracellular matrix. Disruption to the ocular surface and any of these components by injury will trigger a complex cascade of signals to recover all cell types and structures [1]. While mounting evidence has been collected for characterizing the anatomical aspects of nerve recovery, the precise mechanisms underlying successful regeneration and restoration of function are still being elucidated. This review organizes the current understanding of corneal nerve regeneration pathways into early, middle, and late stages, emphasizing the key cellular and molecular events, and discussing the persistent gaps in knowledge regarding anatomical and functional recovery. By highlighting structural repair processes with functional sensory recovery, this review aims to advance the field by synthesizing diverse molecular findings into a unified mechanistic model of corneal nerve regeneration.

## 2. Early Stages: Initial Response, Debris Clearance, and Axon Retraction

Immediately following corneal nerve injury, a cascade of molecular signaling events is triggered to initiate repair and regeneration. Injury triggers ATP release, activating purinergic receptors, and releases neuropeptides such as Substance P and calcitonin gene-related peptide (CGRP), which facilitate epithelial repair and immune cell recruitment. Corneal nociceptors with TRPV1 activation lead to calcium influx, promoting further signaling cascades necessary for early repair. Wallerian degeneration occurs with the damaged axon retracting and preparing the environment for axon regrowth through the extracellular space.

### 2.1. Purinergic and Nociceptor Signaling

Adenosine triphosphate (ATP) release is one of the earliest signaling events following corneal nerve injury, occurring rapidly as damaged epithelial cells and injured neurons release ATP into the extracellular space [13]. This initiates the wound healing process with extracellular ATP activating purinergic receptors, including P2X_3_ on sensory neurons and P2Y_2_/P2Y_4_ on epithelial and immune cells [14]. These receptors act to trigger intracellular calcium mobilization from endoplasmic reticulum stores that spread via jap junctions to synchronize the activity of epithelial cells for their targeted migration towards the wound edge [13,15,16]. ATP release disrupts the local electric fields of the cornea to additionally guide epithelial cell migration [17]. The mobilization of ATP and, subsequently, calcium orchestrates a coordinated response for effective wound healing in the cornea, directly supporting epithelial repair and indirectly supporting nerve regeneration [18].

Polymodal nociceptors (PMNs) populate up to 70% of all cornea nerve types, making them the main detector of pain in the cornea [4,19,20,21]. These neurons, characterized by their responsiveness to chemical, thermal, and mechanical noxious stimuli, prominently express Transient Receptor Potential Vanilloid 1 (TRPV1), which acts as a molecular detector of injury signals and initiates regenerative processes [4,20,22,23]. TRPV1 is a calcium-gated ion channel activated by heat (>37 °C), acidic pH, deleterious mechanical force, and a myriad of endogenous and exogenous agonists like anandamide, capsaicin, and resiniferatoxin [24]. Upon injury, released inflammatory mediators like ATP, protons, bradykinin, and prostaglandins, along with upregulated nerve growth factor, rapidly activate TRPV1 channels in sensory nerve terminals [22,25,26]. Upon activation, TRPV1 binds calmodulin (CaM) and mobilizes internal calcium stores, and promotes cellular processes necessary for corneal nerve regeneration [27]. Activation of TRPV1 triggers the release of Substance P and CGRP from sensory nerve terminals, which in turn send nociceptive signals that actively contribute to the regenerative process [28,29,30,31]. PMNs regulate neuroinflammation, enhance neurotrophic signaling, and facilitate epithelial-nerve crosstalk, emphasizing their dual sensory and regenerative roles during wound healing [28]. In animal models, TRPV1^+^-nerve fibers demonstrate a higher density in the regenerating basal epithelium compared to TRPV1^−^-fibers and maintain comparable terminal lengths in superficial epithelial layers relative to intact corneas, indicating a robust regenerative capacity of TRPV1^+^ -nerve fibers during early healing phases [32]. Animal studies utilizing TRPV1 knockout mice have demonstrated that TRPV1 promotes axon regeneration after injury and modulates corneal nerve sensation, highlighting its role in the regeneration of both structure and function [33,34,35].

### 2.2. Axon Breakdown and Preparation for Regeneration

Peripheral sensory nerves have a great capacity to regenerate, with new growth cones typically developing within hours [36]. In peripheral sensory nerve axons, Wallerian degeneration is a well-described process in which the injured axon is disintegrated distal to the injury site [37]. Within an instant after injury, calcium is introduced to the injury site, and the process of immediate denervation occurs [38]. This includes distal axon fragmentation, disassembly of cytoskeletal elements, and immune infiltration for debris clearance [37,39]. In peripheral nerves, injury signals are packaged into signaling endosomes or bind to importin proteins (notably importin-β1), which are necessary for retrograde injury signaling [40,41]. These complexes recruit dynein, the retrograde motor protein, for transport along microtubules back to the neuron soma [42]. The retrogradely transported complexes activate transcription factors such as c-Jun, ATF3, and CREB, which collectively coordinate to promote axon regeneration [43,44,45,46]. Following injury, axons undergo retraction with a bulb formation, a process mediated by proteases released from Schwann and immune cells, preceding growth cone reformation [47]. In myelinated nerves, Schwann cells activate c-Jun [48,49], secrete matrix metalloproteinases (MMPs) [50], and phagocytose debris [51]. These cells collectively form bands of Büngner, which are columns of aligned Schwann cells and ECM that provide guidance channels for regenerating axons [52]. This process is an important aspect of priming the matrix environment for new axon outgrowth.

Corneal epithelial cells act as surrogate Schwann cells in the process of axon breakdown in cornea injury, actively phagocytosing degenerating intraepithelial corneal nerve fragments within lysosomes [53]. Epithelial cells engulf and digest fragmented axonal material, a process regulated by cell surface proteins such as integrin αvβ5 and syndecan-1 [54]. In the trigeminal ganglion, retrograde transport molecules further modify neuronal repair gene expression, priming the environment for nerve regeneration for subsequent axonal guidance, extension, and reinnervation [55,56,57]. During injury, damaged epithelial and stromal cells release damage-associated molecular patterns (DAMPs) that activate inflammatory cascades and recruit immune cells [58]. Meanwhile, after injury, the basement membrane and ECM are disrupted, exposing axon terminals to stress and activating cell-surface receptors, including transient receptor potential channels and integrins, setting the stage for axon guidance and regeneration [59,60,61].

## 3. Initiation of Regeneration: Extension and Elongation

The following section emphasizes the signaling cascades that become activated for the regeneration of the damaged corneal nerve, typically with signals originating from the corneal epithelium and stroma. The paracrine fashion in which nerves are signaled to regenerate serves as an important and synergistic mechanism in which both the epithelium and corneal nerves regenerate. The focus here will remain on the impact of corneal nerve regeneration and how the coordinated signals promote nerve extension and elongation through a primed environment.

### 3.1. Growth Cone Formation

The formation of a new growth cone at the proximal axon stump after Wallerian degeneration marks the onset of regeneration. This is triggered by intrinsic neuronal pathways (like the upregulation of regeneration-associated genes [RAGs]) and extrinsic factors like neurotrophins or axon-guidance molecules [62,63]. Growth cones, the dynamic structures at the tips of regenerating axons, advance through the corneal stroma by responding to molecular guidance cues. These cues help direct axonal extension and branching, ensuring proper nerve patterning in the cornea [64]. Regenerating subbasal axons form growth cones that extend along the basement membrane, navigating disrupted ECM, laminin from migrating epithelial cells, and dynamic integrin environments as MMPs cleave ECM components during reepithelialization [38,65]. In the cornea, regeneration is accompanied by upregulation of RAGs in both nerves and Schwann-like epithelial-support cells, including growth-associated protein GAP-43, which is expressed even in unwounded cornea, indicating continual remodeling [53,55,56].

### 3.2. Paving the Way: Extracellular Matrix (ECM) Remodeling for Nerve Extension

The ECM is a dynamic and bioactive scaffold that helps facilitate corneal nerve regeneration after injury. It provides both structural support and molecular cues that guide axonal growth, modulate cellular behavior, and regulate the wound healing microenvironment. Key molecules include integrins, laminins, collagens, and fibronectin, which provide directional cues, while MMPs and their TIMPs modulate ECM structural integrity. Integrin-ECM interactions activate focal adhesion kinase (FAK) and YAP/TAZ pathways, influencing axonal adhesion, migration, and survival [50,66]. The corneal ECM is primarily composed of type I collagen, with additional contributions from type IV and XVIII collagens, laminins, fibronectin, and proteoglycans [67]. These molecules are organized into distinct layers in the cornea: the epithelial basement membrane, Bowman’s layer, stroma, and Descemet’s membrane [68,69]. Laminin, in particular, is critical for neurite extension and branching while acting as a substrate for axonal adhesion and migration [70]. Collagen XVIII and laminin-211 are essential for proper axonal organization—loss of these components leads to disorganized nerve regrowth and delayed reinnervation [69,71,72,73].

Corneal nerves and epithelial cells express integrins (notably α3β1, α6β1, and α6β4) that bind ECM laminin and collagen [74,75]. The binding of integrin triggers integrin clustering of the extracellular-binding domain and activation of FAK, which initiates downstream signaling for cytoskeletal reorganization, cell migration, and survival [66,74]. The FAK-Src-YAP/TAZ axis is a central pathway by which corneal cells sense and respond to ECM cues during nerve repair [76]. Activation of this signaling network promotes axon regeneration, epithelial wound healing, and stromal remodeling, all essential for restoring corneal function after injury [77]. Integrin remodeling leads to the formation of focal adhesions and actin polymerization, an essential step for growth cone advancement and axon extension [74].

Following injury, the ECM undergoes rapid remodeling, and MMPs, especially MMP12 and MMP9, are upregulated to degrade damaged ECM, facilitate cell migration, and release sequestered growth factors in a controlled manner [78,79]. This remodeling is necessary for clearing debris and creating a permissive environment for nerve and epithelial regeneration [80,81]. Nerve Growth Factor (NGF)—an essential polypeptide in the survival of neurons—directly stimulates the expression and activation of MMP9 in corneal epithelial cells via TrkA receptor signaling [82,83]. MMP upregulation happens quickly and transiently, with its expression peaking during the early phases of wound healing [82]. By degrading collagen IV and laminin in the basement membrane, MMP9 creates permissive pathways for regenerating nerve fibers to extend through the stroma and reinnervate the corneal surface—an essential aspect for restoring corneal sensation and function. Tissue inhibitors of metalloproteinases (TIMPs) are critical regulators of ECM remodeling in the cornea, balancing the activity of MMPs to ensure proper tissue repair and nerve regeneration after injury. By moderating ECM remodeling, TIMPs influence cellular migration, proliferation, and the ECM environment necessary for effective nerve healing [50,82,84,85]. TIMPs bind to the catalytic sites of MMPs, blocking their proteolytic activity and protecting collagen, laminin, and fibronectin from excessive degradation. This inhibition ensures remodeling is controlled, allowing for the removal of damaged matrix while supporting the arrangement of new ECM necessary for nerve regrowth [84,86]. Ultimately, the ECM modulates activity in the regenerating cornea through sequestration, activation, and inhibition of various growth factors and structural components, coordinating the molecular events required for effective epithelial and nerve repair.

### 3.3. Neurotrophic Factors in Axon Regeneration

In addition to ECM remodeling that paves the way for neuronal outgrowth, neurotrophic factor secretion plays a vital role in orchestrating nerve repair. These specialized growth factors play a role in supporting neuronal survival, guiding axonal regeneration, and modulating neurite architecture. This section will explore the major classes of neurotrophic factors, their mechanisms of action, and their roles in regulating corneal nerve repair.

#### 3.3.1. Docosanoid Synthesis and Initiating the Regeneration Cascade

Upon injury, pigment epithelium-derived factor (PEDF) is secreted from the corneal epithelium and binds to its receptor (PEDF-R), both present in the epithelium and corneal nerves, activating intrinsic calcium-independent phospholipase A2ζ (iPLA2ζ) activity [87,88,89]. This activation triggers the release of docosahexaenoic acid (DHA) from membrane phospholipids for docosanoid synthesis [87,90,91]. The free DHA serves as substrate for lipoxygenase-mediated synthesis of docosanoids, specifically neuroprotectin D1 (NPD1) and the novel resolvin D6 stereoisomer (RvD6si) [92]. NPD1 stimulates the synthesis of brain-derived neurotrophic factor (BDNF), NGF, and the axon growth promoter semaphorin 7A (Sema7A) in corneal epithelial cells and trigeminal ganglia [87]. These factors collectively promote axonal regrowth within the corneal stroma, induce corneal nerve regeneration, enhance wound healing, and restore corneal sensitivity after injury [87,92,93]. RvD6si further enhances neurogenesis via mTORC2 pathway activation while suppressing neuropathic pain markers, supporting growth cone formation and nerve regeneration [92]. This docosanoid-mediated pathway represents a targetable mechanism where topical PEDF plus DHA treatment accelerates wound healing and nerve recovery in mouse and post-surgical models [87,90,94]. The discovery of RvD6si as an active mediator unlocks new avenues for therapies that bypass the need for PEDF-R activation while maintaining nerve regenerative efficacy.

#### 3.3.2. Nerve Growth Factor

Since its discovery in the 1950s, NGF has been recognized as a critical neurotrophin supporting the survival, growth, and regeneration of neurons in both the central and peripheral nervous systems [95]. NGF is rapidly upregulated following injury in the epithelium, with peak expression occurring 24–48 h post-injury at the epithelial-wound edge [93]. NGF binds to TrkA and p75(NTR) receptors on damaged sensory axons, which mediate opposing functions relevant for nerve repair. TrkA activation triggers PI3K/Akt and MAPK/ERK pathways that promote neuronal survival, axonal elongation, and growth cone motility [82,93,96,97,98,99,100]. When NGF binds to p75(NTR), the receptor recruits adaptor proteins that trigger RhoA activation—a small GTPase—which leads to cytoskeletal changes while promoting apoptotic signaling [101]. RhoA activation then facilitates the activation of the JNK (c-Jun N-terminal kinase) pathway, activating apoptotic machinery [102]. The activation of p75(NTR) receptors ultimately leads to a disruption in nerve growth, often accompanied by the induction of apoptosis in damaged axons as a regulatory checkpoint [103,104]. This suggests a balance between TrkA and p75 activation to prevent aberrant nerve re-growth and a mechanism to fine-tune regeneration.

NGF also induces the synthesis of neuropeptides like substance P and CGRP, which promote epithelial healing and stimulate insulin-like growth factor-1 (IGF-1) signaling, further accelerating wound closure [105,106]. Additionally, retrograde NGF transport to the trigeminal ganglia amplifies neurotrophic support signals by delivering growth-promoting cues from peripheral targets back to neuronal cell bodies [107]. Functionally, NGF treatment has been shown to restore corneal nerve density and sensory function in animal and clinical models [93,99,108,109]. NGF released by epithelial cells post-injury binds to TrkA receptors, activating PI3K/PKCε pathways that phosphorylate TRPV1 on sensory neurons, lowering its activation threshold, and enhancing sensitivity, creating a positive feedback loop that promotes nerve regeneration [110,111,112]. This dual action—local epithelial repair and neuronal transcriptional activation—positions NGF as a crucial regulator of both structural and sensory recovery in the cornea [107,113].

#### 3.3.3. Glial Cell-Derived Neurotrophic Factor (GDNF)

GDNF promotes the survival and proliferation of corneal epithelial cells, enhances nerve regeneration within the cornea, and can improve corneal wound healing in both normal and diabetic conditions [93,114]. In both human and animal models, studies utilizing RT-PCR and in situ hybridization demonstrate GDNF mRNA is transcribed in cultured stromal keratocytes but not in corneal epithelial cells, suggesting keratocytes within the stroma are the main producers of GDNF in the cornea [115,116]. Although GDNF is not transcribed in epithelial cells, they do express GDNF receptors (GFRa-1), highlighting the relationship these cells share during cornea wound regeneration [115,117]. After cornea injury, GDNF production and release are increased, with GDNF acting locally on GFRα-1 in the epithelium and sensory neurons [115,118]. Once bound to GFRα-1, GDNF triggers FAK/Pyk2 phosphorylation to promote cytoskeletal reorganization and cell migration. GDNF also activates the epithelial MAPK cascade, leading to the phosphorylation of kinases and transcription factors c-Raf, MEK1, ERK1/2, and Elk-1 to promote gene transcription for wound repair [96,119,120,121]. GDNF signaling activates paxillin phosphorylation to enhance focal adhesion turnover, an essential process for epithelial migration [115]. In primary mouse trigeminal ganglia neurons, GDNF promotes neurite elongation and branching, while its neutralization results in impaired subbasal nerve regeneration [93]. Silk fibroin-based scaffolds containing GDNF have been shown to stimulate epithelial and keratocyte regeneration, also showing an increase in GAP43-positive nerve fibers, highlighting the accommodation of epithelial cell regeneration with an increase in growth cone activity during nerve regeneration [122].

#### 3.3.4. Brain-Derived Neurotrophic Factor (BDNF)

In the peripheral nervous system, Schwann cells surrounding nerves contribute to BDNF synthesis post-injury [123,124]. In the human cornea, BDNF transcription occurs in both the epithelium and stroma, though at lower levels compared to other neurotrophins [117]. After corneal nerve damage, BDNF is primarily secreted by stromal keratocytes, corneal epithelial cells, and fibroblasts [55]. BDNF exerts its effects through binding to the tyrosine kinase receptor TrkB, activating phosphorylation of extracellular signal-regulated kinase 1 (ERK 1) in epithelial cells. ERK signaling promotes neuronal survival and axon growth, facilitating nerve regeneration. BDNF enhances corneal epithelial colony formation, creating a microenvironment conducive to nerve repair through support of the surrounding epithelium [117]. Studies demonstrate that excessive norepinephrine depletes BDNF via β2-adrenergic receptor activation, impairing nerve regeneration and wound healing, underscoring the need for balanced signaling during healing [125].

#### 3.3.5. Neurotrophins-3 and -4

Neurotrophin-3 (NT-3) and neurotrophin-4 (NT-4) are key members of the neurotrophin family, transcribed in the human corneal epithelium, with NT-3 also present in the stroma, while NT-4 expression appears to be restricted to the epithelial layer [117,126]. NT-3 primarily signals through the TrkC receptor, which is expressed on corneal nerves and epithelial cells, while NT-4 binds to the TrkB receptor, shared with BDNF [117]. These receptors activate downstream intracellular ERK1/2 and PI3K/Akt pathways, which promote neuronal survival, neurite outgrowth, and cytoskeletal remodeling. Experimental studies have shown that NT-3 and NT-4 can promote neurite extension and branching in dorsal root ganglia sensory neurons in vitro [126,127]. NT-3 activates ERK-CREB signaling in sensory neurons, facilitating axon growth and bundle formation [128]. Their paracrine action within the corneal epithelium and stroma underscores their importance in sustaining the regenerative capacity of the corneal nerve network following injury.

#### 3.3.6. Ciliary Neurotrophic Factor (CNTF)

CNTF plays an important role in corneal nerve repair by modulating corneal epithelial stem/progenitor cell activity and nerve regeneration. After corneal injury, CNTF is primarily secreted by corneal epithelial stem/progenitor cells, with expression significantly upregulated at both gene and protein levels [129]. CNTF increases colony-forming efficiency 2.5-fold and upregulates stem markers (ΔNP63, Ki-67, ABCG2), demonstrating enhanced epithelial progenitor cell proliferation [130]. CNTF signals through a three-receptor complex comprising CNTFRα, gp130, and LIFRβ, and this binding triggers downstream JAK/STAT3 pathway activation, evidenced by rapid STAT3 phosphorylation (peak at 15–30 min) [130]. CNFT induces Akt phosphorylation, elevating MMP3/MMP14 expression to accelerate epithelial cell migration and wound closure [129]. Topical CNTF elevates nerve fiber density to 93% of baseline within 8 weeks vs. 73% in controls, restoring corneal sensitivity [131]. CNTF’s dual action on epithelial repair and nerve regrowth highlights its importance in corneal nerve regeneration by supporting epithelial wound closure and improving corneal nerve fiber density, along with sensation.

#### 3.3.7. Insulin-like Growth Factor-1

Insulin-like growth factor-1 (IGF-1) is a potent growth factor that plays a multifaceted role in corneal regeneration following injury [132,133]. IGF-1 is secreted by corneal epithelial cells and stromal keratocytes and acts through its receptor, IGF-1R, which is expressed on corneal nerves, epithelial cells, and stromal cells [134,135,136]. Upon activation, IGF-1R initiates downstream signaling cascades such as PI3K/Akt, MAPK/ERK, and p38 MAPK, which collectively promote cell survival, proliferation, migration, and axonal elongation [136,137]. IGF-1 helps preserve corneal stem/progenitor cell markers (e.g., Hes1, Keratin15, p75) and promotes the differentiation of limbal stem cells into corneal epithelial cells, aiding in long-term epithelial and nerve regeneration [138,139]. Additionally, IGF-1 enhances corneal epithelial wound closure and barrier function when combined with substance P by synergistically activating pathways that drive epithelial cell migration and adhesion [106]. IGF-1 also supports the maintenance and differentiation of corneal stem/progenitor cells, facilitating long-term epithelial and nerve regeneration [139]. Beyond its neurotrophic effects, IGF-1 modulates stromal and keratocyte interactions, preserves tissue homeostasis, and prevents pathological neovascularization, thereby maintaining corneal clarity and function [133]. Evidence from both animal models and translational studies demonstrates that IGF-1, alone or in combination with other trophic factors, accelerates healing and restores corneal integrity after injury, underscoring its therapeutic potential in the management of neurotrophic keratopathy and other forms of corneal neuropathy [106,133].

#### 3.3.8. Vascular Endothelial Growth Factor (VEGF)

Traditionally recognized as a master regulator of angiogenesis, VEGF is increasingly acknowledged as a key mediator for corneal nerve regeneration. Following epithelial or stromal injury, the corneal epithelium, stroma, and sensory neurons rapidly increase isoform-specific VEGF production, engaging VEGFR1, VEGFR2, and the co-receptor neuropilin-1 (NRP1) to activate PI3K/Akt and ERK regeneration pathways. These cascades accelerate axon elongation, restore mechanosensation, and promote epithelial repair without necessarily triggering pathological neovascularization [140]. VEGF expression rises markedly within 24 h of injury—both at the mRNA and protein level in the epithelium and stroma—and remains elevated for at least one week [140]. In vivo VEGF treatment after corneal nerve injury significantly increases nerve density post-epithelial recovery, while VEGF inhibitor studies show that neutralizing VEGFR1, VEGFR2, or NRP1 abolishes VEGF-induced neurite elongation in trigeminal ganglia, indicating that coordinated activation of all three receptors is required for proper neurite distribution [140,141]. The co-expression of VEGFR1/2 and NRP1 in the corneal epithelium creates an epithelial “reservoir” that spatially directs VEGF toward regenerating axons [142].

Distinct from VEGF-A, VEGF-B also promotes corneal sensory nerve elongation and branching, enhancing the functional and trophic recovery during wound healing [142]. Mechanistically, VEGF-B enhances the local release of neurotrophic factors, notably increasing pigment epithelium–derived factor (PEDF), which supports neuronal survival, axon outgrowth, and epithelial restoration [143]. Collectively, these findings reposition VEGF from a purely angiogenic regulator to a pivotal ligand in corneal nerve repair and regeneration.

#### 3.3.9. Hepatocyte Growth Factor

Hepatocyte growth factor (HGF) and its receptor c-Met are critical mediators of corneal wound healing and nerve regeneration. Following corneal injury, HGF, primarily secreted by corneal stromal fibroblasts, is rapidly upregulated and signals through c-Met expressed on corneal epithelial, stromal, and endothelial cells, as well as sensory nerves. This interaction begins signaling cascades that promote cell survival, proliferation, migration, and axonal regrowth [144,145]. HGF binding induces c-Met phosphorylation, triggering the MAPK/ERK pathway, to drive epithelial cell proliferation and motility while simultaneously supporting sensory nerve regeneration. In parallel, PI3K/Akt activation enhances cell survival and inhibits apoptosis, preserving the viability of both regenerating nerves and epithelium [146]. Therapeutic enhancement of HGF/c-Met signaling, whether by topical application or gene therapy, accelerates epithelial wound closure and improves sensory recovery, positioning this pathway as a promising target for optimizing corneal nerve repair [147].

## 4. Nerve Branching and Patterning in the Regenerating Cornea

As axons elongate, they form branches to re-innervate the corneal tissue. Branching is regulated by a balance of attractive and repulsive cues, including semaphorins, ephrins, and netrins, which determine the pattern and density of re-innervation. While also known to be involved in the extension of corneal nerves, the following molecules and biophysical phenomena are considered in this section for their contribution to neuron branching and restoring axon density in the regenerated cornea. Table 1 below this section organizes and summarizes the ligands and signals discussed in Section 3 and Section 4.

### 4.1. Axon Guidance Molecules and Corneal Nerve Anatomical Distribution

Axon guidance molecules are fundamental in modulating the precise patterning and regeneration of corneal nerves after injury. These molecules—primarily semaphorins, ephrins, and netrins—interact with their respective receptors to direct axonal pathfinding, branching, and reinnervation, ensuring that regenerating nerves reestablish the highly ordered architecture necessary for corneal transparency and function. These molecules, initially identified for their roles in neural development, exhibit context-dependent functions in adult corneal repair, sometimes promoting regeneration and other times inhibiting it. The following is an overview of key proteins, their receptors, and downstream effectors in corneal nerve architecture.

#### 4.1.1. Semaphorins

Semaphorins are a diverse family of axon guidance cues with context-dependent roles in corneal nerve regeneration and anatomical patterning. Following injury, semaphorins activate receptor-mediated signaling cascades that remodel the cytoskeleton, promote neurite extension, and modulate inflammation, thereby coordinating organized and functional nerve repair in the cornea [148]. Semaphorins act primarily through NRP1 and neuropilin-2 (NRP2), expressed on sensory nerve fibers and trigeminal ganglion neurons. These receptors form complexes with plexin-A family members to transduce intracellular signals [52,148,149]. After corneal injury, semaphorin 3A (Sema3A) and semaphorin 7a (Sema7A) are markedly upregulated in the corneal epithelium and stroma, with a 3- to 10-fold increase in expression within the first 24 h [52,150].

Sema3A, produced by both epithelial cells and stromal keratocytes (with secretion concentrated in the regenerating epithelium), is classically known as a growth cone inhibitor via NRP1/plexin signaling during development [148,150,151]. However, in the injured adult cornea, Sema3A can shift to a growth-promoting role. In trigeminal neurons of adult mice, Sema3A activates Rho GTPase–dependent cytoskeletal remodeling, enhancing neurite growth and branching in a manner comparable to NGF [148,150]. This functional switch underscores Sema3A’s importance in structural nerve regeneration.

Sema7A, upregulated in both corneal epithelial cells and tears after injury, signals through β1 integrin and focal adhesion kinase (FAK) to promote axonal growth and recruit inflammatory cells [91,148,152]. In mouse models, Sema7A increases neurite length and its expression near regenerating nerve branches and in the stroma suggests a role in guiding axons along defined stromal and epithelial trajectories. This contributes to the restoration of nerve density and the characteristic radial and centripetal organization of corneal nerves [152].

#### 4.1.2. Ephrins

Ephrin receptors (Ephs) are the largest family of tyrosine kinase receptors, and Eph/ephrin signaling plays a central role in spatial patterning and guidance of corneal nerve reinnervation [153]. Eph receptors (notably EphA2, EphA3, EphB1, and EphB4) and their ligands, ephrins (ephrin-A1, ephrin-B1, and ephrin-B2), are expressed in the corneal epithelium, stroma, and trigeminal nerves, with their expression patterns dynamically regulated after injury [154,155].

EphA3, expressed in the corneal epithelium and in trigeminal ganglion neurons, mediates axon guidance by binding to EphA4 on adjacent nerve fibers, contributing to precise nerve patterning during nerve regeneration [153]. Eph binding on adjacent neurons triggers autophosphorylation of the receptor’s intracellular tyrosine residues, initiating a cascade of downstream signaling events [156]. EphA2/ephrin-A1 signaling primarily activates PI3K/Akt and ERK1/2 pathways, whereas EphB/ephrin-B signaling engages Rho family GTPases [154,157]. Together, these cascades regulate cytoskeletal remodeling, cell migration, and axon guidance, ensuring effective corneal nerve repair.

#### 4.1.3. Netrins

Netrins are a family of axon guidance proteins that play a pivotal role in corneal nerve regeneration and epithelial wound healing following injury. The netrin receptors DCC, Neogenin, and UNC5H1 are expressed in the corneal epithelium and interact with Netrin-1 and Netrin-4, while Neogenin and α6β1 integrin are key mediators of Netrin-4-induced neurite growth and epithelial migration [52]. Under baseline conditions, netrin-4 is highly expressed in the corneal epithelium and trigeminal ganglia [158]. Upon binding to Neogenin or α6β1 integrin on corneal nerves, netrin-4 activates the PI3K/Akt pathway, promoting neuronal survival, neurite extension, branching, and cytoskeletal reorganization [159]. Concurrent ERK1/2 activation by netrin-4 binding supports migration and proliferation in neurons and epithelial cells, accelerating both wound closure and nerve regeneration [160]. Netrin-4 can also activate FAK via α6β1 integrin binding, further enhancing cytoskeletal remodeling and cell motility [159].

In vitro, netrin-4 induces neurite growth in cultured trigeminal ganglion neurons to a greater extent than NGF, and in vivo, topical netrin-4 accelerates corneal epithelial healing and nerve regeneration in mouse injury models. In cultured human corneal epithelial cells, netrin-4 accelerates wound closure by enhancing cell migration and proliferation [158]. In vitro treatment of netrin-4 potently induces neurite growth in cultured trigeminal ganglia cells as compared to NGF-treated cells. Additionally, in vivo treatment of mice with epithelial wounds by netrin-4 displayed accelerated epithelium healing and nerve regeneration [158]. Netrin-1 leads to axonal attraction or repulsion depending on whether it binds to DCC or UNC5, respectively [161,162]. It can upregulate epidermal growth factor (EGF) expression and activate ERK and EGFR signaling, driving epithelial cell proliferation and migration [163]. Netrin-1 induces the formation of UNC5B-DCC receptor complexes, which can modulate cytoskeletal dynamics and promote axon guidance and extension [164]. Collectively, netrins, particularly netrin-4 and netrin-1, are crucial regulators of corneal nerve and epithelial regeneration after injury. By engaging specific receptors and activating intracellular cascades such as PI3K/Akt, ERK1/2, and FAK pathways, they drive neurite extension, cytoskeletal reorganization, and epithelial wound closure. These properties position netrins as promising targets for enhancing corneal nerve repair after injury.

### 4.2. Electrical Fields in Corneal Nerve Regeneration

Although distinct from intracellular signaling cascades, endogenous electrical fields (EFs) are foundational biophysical cues that coordinate epithelial wound healing and corneal nerve growth cone regeneration [165,166]. Under physiological conditions, the cornea maintains an internally positive transcorneal potential difference (TCPD) of approximately +40 mV under normal conditions, established by active transport of Cl^−^ outward and Na^+^ inward via ion channels, Na^+^/K^+^ ATPases, and tight junctions within the corneal epithelium [17,166,167,168,169,170]. Following injury, the TCPD collapses instantaneously at the center of the wound, while remaining intact in adjacent, uninjured tissue, creating a voltage gradient of ~40 mV/mm. This generates a lateral EF oriented towards the wound, with the center of the wound acting as a negatively charged cathode. This injury-induced EF serves as a strong guidance cue, directing both epithelial cell migration and axonal sprouting toward the wound center [171,172]. Beyond directional guidance, EFs also increase the overall number of newly sprouting nerves, with stronger fields near the wound edge correlating with higher sprouting rates [17,166,173]. Pharmacological studies reveal that inhibition of ATP signaling via pannexin-1 and purigenic P2X receptor blockade significantly impairs the migration of corneal epithelial cells and dwindles nerve sprouting in vitro, while exogenous ATP enhances EF strength and increases nerve density. ATPase inhibitors further amplify EF effects, promoting regeneration while enabling corneal epithelial cells to provide trophic support during migration [17]. Thus, EFs—generated and modulated by ion transport and ATP signaling—are essential biophysical cues in the orchestration of structural and functional repair of corneal nerves, emphasizing the dynamic interplay between epithelial and neuronal components necessary for successful regeneration.

### 4.3. Mechanical Interactions

The mechanical stiffness of the cornea is a critical determinant of nerve regeneration and the restoration of normal nerve patterning after injury. Changes in tissue stiffness, primarily resulting from ECM remodeling and cellular differentiation, directly influence both the capacity for axonal regrowth and the spatial distribution of regenerating nerves. After corneal injury, keratocytes in the stroma differentiate into fibroblasts and myofibroblasts, which deposit increased amounts of collagen and contractile proteins. This process leads to a stiffer ECM, which can act as a physical and biochemical barrier to regenerating nerves [174,175,176,177]. The differentiation of keratocytes and the resulting ECM stiffening are mediated in part by growth factors such as transforming growth factor-β1 (TGF-β1), which is upregulated after injury [178]. Strategies that reduce ECM stiffness—such as inhibiting myofibroblast differentiation or blocking TGF-β1 signaling—have been shown to enhance nerve regeneration and promote the restoration of normal nerve patterns [174,175]. The mechanical stiffness of the regenerating cornea not only affects the efficiency of nerve regrowth but is also fundamental in guiding the re-establishment of the cornea’s intricate nerve architecture. Therapeutic approaches targeting ECM remodeling and tissue biomechanics hold promise for optimizing both the anatomical and functional outcomes of corneal nerve regeneration.

**Table 1 cells-14-01322-t001:** Growth Factors and Intracellular Signaling Cascades in Corneal Nerve Regeneration.

Ligand	Receptor	Key Intracellular Pathways	Functional Outcome of Ligand
NGF	TrkA p75	PI3K/Akt, MAPK/ERK, PLCγ JNK, RhoA pathway	Promotes neurite outgrowth, survival, and epithelial healing Promotes apoptosis, restrict excessive proliferation, inhibit growth cone
BDNF	TrkB	PI3K/Akt, MAPK/ERK	Enhances neuronal survival, axon elongation, and epithelial repair
CNTF	CNTFRα	JAK/STAT3	Supports axon regeneration, epithelial healing
GDNF	GFRα1/RET	PI3K/Akt, MAPK/ERK	Promotes axonal regrowth, neuronal survival, neurite elongation
NT-3	TrkC	PI3K/Akt, MAPK/ERK	Supports sensory neuron survival and axonal outgrowth
NT-4	TrkB	PI3K/Akt, MAPK/ERK	Promotes neurite extension, sensory neuron survival
Sema3A	Neuropilin-1/ Plexin-A	Rho GTPases	Regulates cytoskeletal remodeling, promotes or inhibits neurite growth depending on context
Sema7A	β1-integrin	FAK, MAPK/ERK	Enhances axonal growth, inflammatory cell recruitment
Ephrins	Eph receptors (EphA2, EphA3, EphB1, EphB4)	PI3K/Akt, MAPK/ERK, Rho GTPases	Directs axon guidance, patterning, and branching
Netrin-1	DCC, UNC5B	PI3K/Akt, MAPK/ERK,	Promotes axon guidance, cytoskeletal reorganization, epithelial wound healing
Netrin-4	Neogenin, α6β1-integrin	PI3K/Akt, MAPK/ERK, FAK	Enhances neurite extension, epithelial cell migration and proliferation
TGF-β1	TGF-β receptors (also epithelial cells, stromal cells)	SMAD, Rho/ROCK	Modulates ECM remodeling, can inhibit nerve regeneration if persistent
ATP	P2X/P2Y receptors (also epithelial cells)	Calcium influx, purinergic signaling	Initiates wound healing, supports nerve sprouting
VEGF	VEGFR-1/VEGFR-2 (also on endothelial cells)	PI3K/Akt, MAPK/ERK	Promotes nerve regeneration, angiogenesis, supports epithelial healing
PEDF	PEDF-R (also epithelial cells)	PI3K/Akt, anti-angiogenic signaling	Neuroprotective, anti-angiogenic, supports nerve survival
Inflammatory Mediators	TRPV1 channels	Ca^2^^+^ influx, PI3K/Akt, ERK1/2, neuropeptide release (Substance P, CGRP), upregulation of IL-6, TGF-β1	Promotes epithelial and stromal healing, enhances nerve regeneration, modulates neuroimmune response, maintains nerve density and patterning, supports functional recovery

## 5. Emerging Regulators of Corneal Nerve Repair

While classical neurotrophic factors and axon guidance molecules remain central to corneal nerve regeneration, recent research has uncovered additional regulators of control that refine and enhance the regenerative process. MicroRNAs (miRNAs), exosomes, and other extracellular vesicles have emerged as modulators of gene expression, intercellular communication, and microenvironmental signaling, influencing both the anatomical regrowth and functional restoration of corneal nerves. These regulators interact with established canonical pathways, composing complex molecular networks that support neuronal survival, axonal extension, and epithelial integrity after injury. Understanding these modulators offers promising therapeutic avenues to improve outcomes in corneal nerve repair.

### 5.1. Exosomes and Extracellular Vesicles

Exosomes are nanoscale extracellular vesicles (EVs) secreted by diverse cell types, acting as carriers of proteins, lipids, and nucleic acids—including miRNAs—to mediate intercellular communication and coordinate regenerative processes essential for nerve repair [179,180]. By delivering their cargo directly to target cells, exosomes influence gene expression and signaling cascades that drive neurite outgrowth, cell migration, and tissue remodeling [180,181,182]. Given their minimal immunogenicity and potential to be engineered for targeted therapy, exosomes are being actively explored as cell-free therapeutic agents for corneal nerve injuries [183]. In some studies, stem cell–derived exosomes have outperformed classical neurotrophic factors by providing a multitarget approach—modulating immune responses, promoting anti-apoptotic effects, and supporting both epithelial and neuronal repair [180,182,184].

Exosomes derived from mesenchymal stem cells (MSCs) suppress inflammation and fibrosis in the cornea [181,184,185,186]. Exosomes from bone marrow MSCs (BM-MSCs) and corneal MSCs (cMSCs) have demonstrated a significant neurotrophic role, with increased neurite growth in corneal sensory neurons following injury. In vitro, harvested murine TG neurons treated with BM-MSC exosomes showed greater neurite length and complexity compared to controls [187]. The exosome’s unique regenerative potential lies in the capacity to orchestrate complex repair processes, making it a promising frontier for future clinical interventions in ocular surface disease and nerve regeneration [179]. There is currently limited information on how exosomes directly affect corneal nerve regeneration, but several reviews have examined the role of exosomes in ocular surface disease recovery and offer promising insight into the mechanistic opportunities exosome therapy provides [179,180,188].

### 5.2. MicroRNAs (miRNAs)

MicroRNAs (miRNAs) are short, non-coding RNAs that regulate gene expression post-transcriptionally and have emerged as critical modulators of corneal nerve regeneration. By fine-tuning the expression of genes involved in oxidative stress, inflammation, neuronal survival, and axonal growth, specific miRNAs can either promote or hinder repair. The following examples illustrate some of the most conclusive evidence linking miRNAs to corneal nerve regeneration.

#### 5.2.1. MicroRNA-182

MicroRNA-182 has emerged as a modulator of corneal nerve axon regeneration, particularly due to its ability to modulate oxidative stress and support axonal outgrowth. Its principal target is NADPH oxidase 4 (NOX4), an enzyme that generates reactive oxygen species linked to neuronal injury and impaired regeneration [189]. In animal models of diabetic corneal neuropathy, upregulation of miR-182 suppresses NOX4 expression, decreases oxidative damage, and thus creates a more permissive environment for neurite outgrowth. This mechanism results in enhanced regrowth of sensory nerve fibers and improved functional recovery of corneal sensation, even under persistent hyperglycemic stress [190]. This positions microRNA-182 as a modulator of both structural and functional regeneration of corneal nerves.

#### 5.2.2. MicroRNA-223-5p

In diabetic corneal nerve injury, miR-223-5p acts as a negative regulator of both epithelial and nerve repair. Elevated miR-223-5p directly suppresses Hpgds (hematopoietic prostaglandin D synthase), an enzyme that mediates anti-inflammatory signaling and supports tissue regeneration. This suppression increases local inflammation, disrupts epithelial healing, and impairs sensory nerve regrowth. Inhibition of miR-223-5p in diabetic mice restores Hpgds expression, leading to reduced inflammation and the upregulation of genes involved in neuronal growth and epithelial repair. As a result, blocking miR-223-5p promotes both anatomical and functional regeneration of the cornea—improving nerve density and sensation in a hyperglycemic state. These findings identify miR-223-5p as a promising molecular target for mitigating diabetic corneal neuropathy and promoting neuronal repair [191]. Its inhibition restores both nerve structure (density, axonal outgrowth) and function (sensation), making it a dual structural–functional regulator.

#### 5.2.3. MicroRNA-181a

The inhibition of miR-181a has been shown to markedly enhance sensory corneal nerve regeneration, particularly in diabetic models where nerve repair is otherwise impaired. Experimental studies demonstrate that suppressing miR-181a using targeted inhibitors promotes axonal outgrowth in TG neurons and accelerates both corneal epithelial healing and nerve fiber regeneration in vivo [192]. The principal molecular targets of miR-181a in corneal nerve regeneration are ATG5 (autophagy-related protein 5) and Bcl-2 (anti-apoptotic protein). Inhibition of miR-181a in diabetic models leads to increased expression of ATG5 and Bcl-2, enhancing autophagy after damage and neuronal survival, and promoting corneal nerve density and epithelial repair [192].

#### 5.2.4. MicroRNA-183/96/182 Cluster

The miR-183/96/182 cluster (miR-183C) governs both structural and functional aspects of corneal nerve regeneration—structurally by controlling axon density and neuronal architecture, and functionally by modulating sensory receptor expression and tear production to sustain corneal homeostasis [193]. The principal molecular target for this cluster is Cx3cl1 (fractalkine), a chemokine involved in neuron-immune cell signaling; miR-183C represses Cx3cl1 expression in trigeminal ganglion neurons, which subsequently shapes corneal nerve density and the number of resident myeloid immune cells. Knockout mouse models show that loss of this cluster significantly reduces corneal sensory nerve density, sensation, and tear secretion. Functionally, inhibition of the cluster decreases TRPV1 and SP precursor gene expression, indicating a role in sensory modulation [194]. These effects are mediated through cell-type-specific target genes in sensory neurons, underscoring the cluster’s role in maintaining the neural machinery required for normal ocular surface physiology [193,195].

#### 5.2.5. MicroRNA-432-5p

Exosome-loaded thermosensitive hydrogels containing exosomes from iPSC-derived MSCs have been shown to enhance corneal epithelium and stroma regeneration after injury. Exosomal microRNA miR-432-5p was crucial for suppressing extracellular matrix deposition and promoting nerve recovery, while also preventing fibrosis and keratitis in animal models [179,196]. The principal target of miR-432-5p for preventing fibrosis includes suppression of translocation-associated membrane protein 2 (TRAM2). TRAM2 is a regulator of collagen biosynthesis, and its suppression by miR-432-5p reduces the deposition of ECM, preventing scar formation and promoting functional corneal nerve recovery [196].

#### 5.2.6. MicroRNA-24-3p

In corneal nerve injury and fibrosis prevention, miRNA 24-3p targets multiple genes associated with cell migration, wound healing, and ECM regulation. Recent studies indicate miRNA-24-3p (from adipose-derived MSC exosomes) upregulates Rho GTPases, FGFR signaling, CCN family proteins, EGFR, and MMP9, which are all involved in the regulation of epithelial cell migration and maturation, ECM breakdown, and anti-fibrotic signaling [197]. Through these gene targets, miRNA 24-3p enhances the healing of corneal epithelial defects, suppresses stromal fibrosis, and supports axonal regrowth.

#### 5.2.7. Neurotrophin Modulation of MicroRNA

Emerging studies reveal that neurotrophic factors modulate microRNA expression after injury. NGF treatment of corneal epithelial cells has been shown to significantly modulate the expression of 21 distinct microRNAs [198]. These NGF-induced miRNAs regulate key genes involved in neurotrophic signaling pathways, such as PI3K, AKT, MAPK, KRAS, BRAF, RhoA, Cdc42, Rac1, Bax, Bcl-2, and FasL—affecting cell proliferation, survival, growth, and apoptosis in the corneal epithelium. By altering the miRNA signature, NGF not only promotes neuronal survival and axonal extension in the nerves but also orchestrates a larger regenerative response in the corneal epithelium, supporting both structural repair and functional recovery of corneal sensory nerves [198,199].

By identifying and targeting specific miRNAs and their gene targets, researchers have uncovered novel pathways that regulate both anatomical nerve regrowth and the restoration of sensory function in corneal nerves. This insight into microRNA-driven regulation opens the door for innovative, precision, or combination therapies that address the barriers to nerve regeneration and functional recovery in the cornea. There are several review papers examining the role of miRNAs in ocular surface diseases more broadly, beyond corneal nerve regeneration, for the reader to explore [200,201,202]. Future research directions may choose to focus on the transcriptomic profile of specific injury models to specifically examine the neurotrophic factor genes, axon guidance molecules, matrix remodeling, and microRNAs that regulate regeneration. Highlighting these neurotrophin–miRNA interactions underscores a mechanistic bridge between classical growth factor biology and post-transcriptional gene regulation, offering a unique conceptual framework for developing targeted therapies in corneal nerve regeneration.

## 6. Outcomes of Regeneration and Future Directions

Corneal nerve regeneration involves dozens of signaling pathways that operate simultaneously to restore both structure and function after injury. In Figure 1, we depict an overview (non-exhaustive) of key neurotrophic signaling pathways supporting corneal nerve repair. This review has integrated established anatomical evidence with emerging data on functional recovery, emphasizing that structural repair alone does not guarantee sensory restoration.

Aberrant nerve regrowth, altered sensory receptor expression, or poor integration into higher neural circuits can yield anatomically present but functionally deficient nerves. Current metrics such as nerve density do not capture these deficits. A major challenge for researchers and clinicians includes correlating structural and functional recovery with different corneal nerve injuries, which yield varied regenerative outcomes [203]. Context-dependent molecular pathways activated during neuron regeneration have been demonstrated in animal models utilizing axotomy (recapitulating penetrating keratoplasty) and corneal abrasion (recapitulating photorefractive keratitis) to exhibit activation of distinct signaling cascades and gene expression patterns [203]. This brings an additional challenge of varied regeneration pathways to consider when looking to treat specific types of corneal injuries and managing the functional deficits that may arise.

In some instances, regenerated corneal nerves fail to fully restore the original, organized whorl-like architecture of the subbasal nerve plexus. Even when regrowth appears robust, abnormal or incomplete patterning can impair function [1,6,12,204]. The local microenvironment, particularly the presence of myofibroblasts and scar tissue, can further inhibit nerve regeneration or misdirect nerve pathfinding, with factors such as TGF-β1 secreted by myofibroblasts actively suppressing neurite outgrowth and delaying functional recovery [174].

This mismatch between structure and function is well documented. After Laser-Assisted In Situ Keratomileusis (LASIK), case studies show a dissociation between anatomical and functional recovery of corneal nerves in these patients. Immediately following LASIK, corneal sensitivity drops substantially, showing a near-total absence or fragmentation of corneal nerve fiber bundles (CNFBs) in the central cornea. In subsequent months, anatomical regrowth of subbasal nerves occurs, yet even at six months post-procedure, unconnected nerve fibers were observed in the central cornea of these patients [205]. In some cases, locations of the cornea missing healed nerve structures demonstrate reduced sensitivity, yet some sensory function can return even when anatomical reinnervation is lacking [205,206]. These findings indicate that morphological abnormalities in subbasal nerve structure may persist well beyond the apparent resolution of sensory loss. This mismatch underscores that the return of corneal sensation is not solely dependent on the full architectural restoration of the subbasal nerve plexus, and the processes of restoring corneal nerve function remain less understood.

Persistent ocular pain or reduced corneal sensation remain clinical challenges, raising the key question—how can we ensure regenerated nerves function as intended? While neurotrophic factors like NGF and BDNF promote survival and elongation, the molecular orchestration of axon guidance and functional integration is still incompletely understood, particularly as emerging regulators such as microRNAs and extracellular vesicles add new layers of complexity. Neurotrophic keratitis (NK) underscores the clinical stakes of impaired corneal regeneration. Loss of sensory input deprives the epithelium of trophic support, hindering repair, while impaired nerve function abolishes protective reflexes (including blink and tear secretion), increasing susceptibility to injury, infection, and vision loss NK [100,207,208]. This pathophysiology highlights both the critical need for strategies that restore corneal function and the importance of molecular therapies targeting neurotrophic signaling for effective regeneration in NK. Functional recovery is measured clinically by corneal sensation tests and the restoration of protective reflexes such as blinking and tear production [1,209,210]. This recovery is crucial for preventing NK and maintaining corneal epithelial integrity. In patients who received minimally invasive corneal neurotization (the surgical implant of healthy nerve donor tissue), corneal sensation improved significantly only after 18 months, despite earlier anatomical nerve fiber increases [211]. Studies have shown that after corneal injury or surgical interventions, nerve fibers can regenerate and repopulate the corneal stroma and subbasal plexus over months to years [1,211,212]. This suggests a temporal disparity in structural-to-functional recovery where the resolution of anatomical nerve regeneration does not guarantee restored function [211]. The discrepancy between nerve structure and function highlights the need for standardized, objective, and sensitive approaches to better correlate anatomical and functional outcomes [4,21,22]. Structural regeneration can be observed through imaging techniques such as in vivo confocal microscopy (IVCM) and optical coherence tomography, which quantify nerve fiber density, branching, and morphology. However, morphological improvements to damaged nerves may not translate into full functional restoration of corneal sensation and reflexes. Functional assessments like aesthesiometry are subjective and can vary between patients and examiners, limiting the clinical interpretation of these measurements [213].

Functional molecular mechanisms, including ion channel remodeling, are increasingly recognized as key determinants of functional recovery and are discussed in Section 6.1. For review papers that focus more in-depth on examining the functional outcomes of corneal nerve injuries, there are several excellent review papers available that delve into the issues facing corneal nerve regeneration and the consequences of incomplete recovery [1,4,214,215].

### 6.1. Ion Channels in Corneal (Dys)Function

A vital aspect of corneal nerve regeneration includes the ion channels involved in sensory detection and transduction. When nerves regenerate, the mere restoration of their architecture does not guarantee normal corneal sensation or pain modulation. Functional outcomes like pain relief, sensitivity normalization, and restoration of protective reflexes are tightly linked to the appropriate expression, trafficking, and modulation of sensory ion channels—including TRPV1, TRPM8, Piezo2, ASICs, and voltage-gated sodium channels—that govern neuronal excitability and signal transmission. Research has demonstrated that corneal nerve injury causes specific changes in ion channel expression, directly influencing corneal sensation, pain, and functional recovery.

TRPV1 is the main nociceptor in the cornea and is considered the ion channel responsible for pain perception [4,19]. TRPV1 overactivation is linked to pain and degeneration in murine models of dry-eye disease (DED). Wild-type mice exhibited increased corneal sensitivity to capsaicin (TRPV1 agonist) and greater signs of spontaneous pain during DED, while TRPV1-deficient mice were protected from nerve dysfunction and did not develop pain or degeneration [5]. Chronic tear deficiency via lacrimal gland excision (as a model of DED) leads to sensitization of TRPV1-mediated responses through changes in TRPV1 protein expression and its phosphorylation, indicating post-translational modification (phosphorylation at S801) may play a role in the sensitization process [216]. Although TRPV1 antagonists have been developed for ocular pain and dry-eye disease, many first-generation drugs failed in clinical trials due to systemic side effects, including impaired heat perception and hyperthermia [217].

Transient receptor potential cation channel subfamily M member 8 (TRPM8), known for cold and menthol sensing, is upregulated after corneal nerve injury. Research shows increased TRPM8 expression and function in injured trigeminal ganglion cold-sensitive neurons, leading to abnormal cold sensation, ongoing firing, and increased tear production, which are all core features of post-injury dry eye discomfort [203,218,219]. Piezo2 channels, expressed in mechanonociceptors and polymodal nociceptors, may undergo micro-injuries in dry-eye or neuropathic pain, becoming “leaky” and disrupting normal neuronal signaling, contributing to maladaptive sensitization and persistent pain [220]. Acid-sensing ion channels (ASICs), particularly ASIC1a and ASIC3, respond to moderate acidification (pH ~6.6) during inflammation or injury. Blockade of ASIC3 in corneal nerve injury or allergic keratoconjunctivitis models reduces nociceptor sensitization and pain behaviors, whereas activation exacerbates pain responses [219,221,222].

Once sensory information reaches a detecting ion channel, a neuron depolarizes, and voltage-gated sodium channels along the axon propagate the afferent information to the trigeminal ganglia soma [223]. Nav1.7 is critical for pain signaling in peripheral sensory neurons, including the trigeminal ganglion neurons that innervate the cornea. Injury, such as axon transection during surgery, increases Nav1.7 expression and can heighten neuronal excitability, contributing to persistent pain [224]. A specific gain-of-function mutation (P610T) in Nav1.7 was found in patients with ongoing ocular pain after corneal surgery. This mutation impairs slow inactivation, resulting in hyperexcitable neurons and chronic pain despite anatomical regeneration [225]. Additionally, Nav1.8 expression patterns can change after injury, with redistribution or increased activation in corneal sensory neurons contributing to ongoing pain in dry eye and nerve injury optogenetic animal models [226].

This evidence of altered ion channel expression highlights the need for future therapeutic strategies to consider not only structural regeneration but also the function of sensory ion channels. Channels such as TRPV1 are critical for sensory signal transmission and pain perception, and changes in their expression, activity, or post-translational modification can profoundly impact functional recovery, even when anatomical regrowth appears successful. Recognition that functional restoration depends on more than structural repair is driving strategies to target molecules and pathways that support both axon regrowth and accurate signal transmission, including pain modulation [215,227,228]. These findings underscore the importance of addressing molecular determinants of neuronal excitability. Ion channel modulation is emerging as a key therapeutic target, potentially enabling regenerated nerves to transmit afferent signals while preventing chronic pain. Understanding the mechanisms behind sensory nerve signaling is therefore critical for advancing corneal nerve regeneration and maintaining ocular health. Expert reviews provide further detail on the roles of these channels in corneal sensory detection [4,21,222]. Overall, ion channels are central elements of corneal nerve function, influencing sensation, pain, and protective reflexes. Effective regeneration strategies must therefore address not only structural regrowth but also the proper expression, modulation, and integration of these channels to ensure functional recovery and maintain ocular health.

## 7. Conclusions

The cornea relies on a tightly regulated interplay between neuronal, epithelial, and ECM components to restore both structure and function after damage. The early injury signals—such as ATP release, calcium influx, and the generation of endogenous electrical fields (EFs)—set the stage for regeneration. These signals not only coordinate rapid epithelial migration but also activate axonal guidance cues and initiate retrograde transport mechanisms. The retrograde movement of injury signals to the TG upregulates a suite of regeneration-associated genes, which collectively prime sensory neurons for axonal outgrowth and target reinnervation. This dynamic communication between the periphery and the neuronal soma is essential for a successful regenerative response. Neurotrophic factors are shown to be indispensable for both neuronal survival and axonal regrowth. Their upregulation after injury, and the subsequent activation of receptors, trigger intracellular cascades that drive nerve regeneration and support epithelial healing. Axon guidance molecules provide spatial cues that may help direct regenerating axons toward their targets, ensuring proper nerve patterning and functional restoration. Notably, the context-dependent roles of these molecules, sometimes acting as attractants and at other times as repellents, underscore the complexity of the regenerative microenvironment.

The ECM acts as a central regulator of corneal nerve repair, not only providing structural scaffolding but also modulating the availability and activity of growth factors. Integrin-mediated signaling through local pathways translates mechanical and biochemical cues from the ECM into cellular responses that promote axon extension, migration, and survival. The balance between ECM degradation (via MMPs) and protection (via TIMPs) is crucial for creating a permissive environment for nerve regeneration while preventing excessive fibrosis or scarring. Beyond these classical mechanisms, emerging regulators such as exosomes and microRNAs are reshaping our understanding of corneal nerve repair. Exosomes serve as vehicles for intercellular communication, delivering miRNAs and bioactive molecules that modulate inflammation, enhance axonal growth, and promote epithelial-nerve crosstalk during regeneration. miRNAs, in turn, fine-tune gene expression within sensory neurons and epithelial cells, orchestrating pathways that support neurite extension, neuronal survival, and epithelial wound healing. Together, these emerging regulators add additional layers of control that may be harnessed for therapeutic intervention.

Importantly, functional recovery of corneal nerves depends not only on structural regrowth but also on the proper expression and modulation of sensory ion channels. Channels such as TRPV1, TRPM8, Piezo, ASICs, and voltage-gated sodium channels govern neuronal excitability, pain perception, and protective reflexes. Altered expression, trafficking, or post-translational modifications of these channels can result in persistent pain, abnormal sensations, or incomplete reflex restoration—even when anatomical regeneration appears complete. Integrating ion channel function into regenerative strategies is therefore critical to ensure that regenerated nerves can transmit afferent signals accurately, maintain ocular surface homeostasis, and restore protective reflexes.

Corneal nerve regeneration is a complex, multi-layered process essential for restoring the ocular surface. This review integrates classical neurotrophic signaling with emerging regulators like microRNAs, extracellular vesicles, and ion channel modulation, framing corneal nerve repair as both a structural and functional endeavor. Viewing regeneration through this integrated lens explains why some nerves regrow yet remain non-functional and points toward therapies that combine structural restoration with precise functional tuning. While advances in transcriptomics, proteomics, and microscopy have improved our understanding of anatomical regeneration, significant gaps remain in correlating structural repair with full sensory recovery. Future research will critically rely on approaches that consider both anatomical and functional outcomes—including ion channel expression and activity—to achieve true corneal nerve restoration.

## Figures and Tables

**Figure 1 cells-14-01322-f001:**
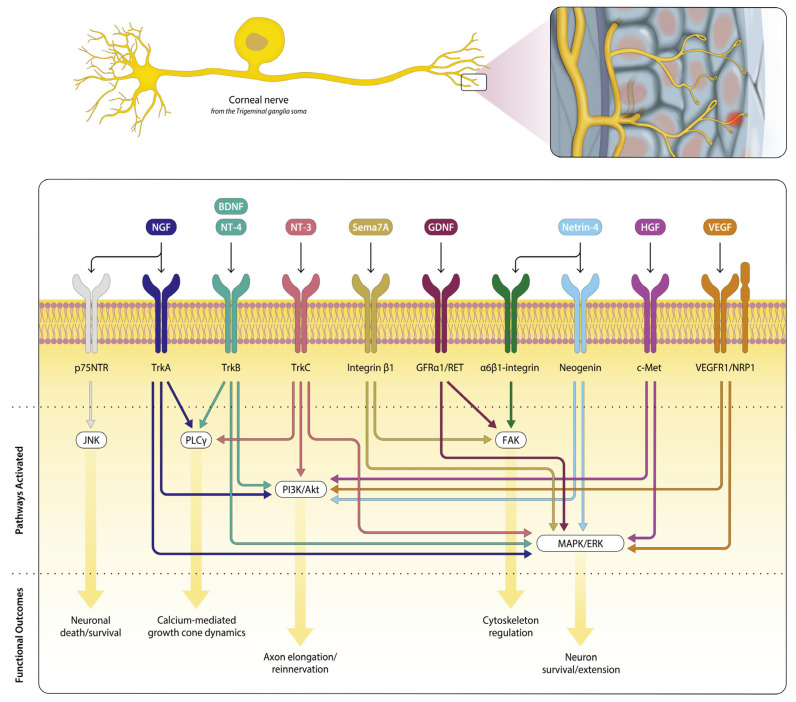
Corneal Nerve Intracellular Signaling in the Regenerating Sensory Neuron.

## Data Availability

No new data were created or analyzed in this study.

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
