# Peer review of "Mechanisms of Corneal Nerve Regeneration: Examining Molecular Regulators"

_cells, 2025, doi:10.3390/cells14171322_

Round 1

Reviewer 1 Report

Comments and Suggestions for Authors

Mechanisms underlying corneal nerve regeneration has been extensively discussed.

The present review is well written and organized.

  1. Although the authors are well experienced in the field, they need to elucidate the added value of their review in respect to additional reviews in the same field.
  2. The authors also need to emphasize their contribution to the field and how their findings might help to understand the mechanisms underlying corneal nerve regeneration.
  3. How future research might help to elucidate all aspects underlying corneal nerve repair should be better discussed.

Reviewer 2 Report

Comments and Suggestions for Authors
  1. The review is comprehensive and important. This reviewer only has some minor issues with the text:
  2. Please change the noun to adjective: e.g., cornea to corneal, where appropriate. For instance, lines 29 and 32.
  3. Laminin-2 is now called laminin-211 (line 169).
  4. Line 171. Please change α3β1, α6β1, and β1 to α3β1, α6β1, and α6β4.
  5. Line 235. The authors might like to mention that NGF receptors may have opposing functions during wound healing (see https://pmc.ncbi.nlm.nih.gov/articles/PMC8312850)
  6. In the miR section, it is important to disclose the targets of the functional miRs. If these targets are unknown, please list this as a limitation of the relevant studies, requiring further in-depth experimentation.
  7. More emphasis on neurotrophic keratopathy and diabetic keratopathy may be advantageous, with pertinent citations.
  8. There are some more recently tested molecules that may be beneficial for corneal regeneration. Also, some general papers may need to be cited:

https://pubmed.ncbi.nlm.nih.gov/28966630/

https://www.nature.com/articles/s41598-025-10434-y
